# Genetic Diversity Analysis Reveals Potential of the Green Peach Aphid (*Myzus persicae*) Resistance in Ethiopian Mustard

**DOI:** 10.3390/ijms232213736

**Published:** 2022-11-08

**Authors:** Fangyuan Zhou, Chaoquan Chen, Lijun Kong, Shenglanjia Liu, Kun Zhao, Yi Zhang, Tong Zhao, Kaiwen Liu, Xiaolin Yu

**Affiliations:** 1Department of Horticulture, College of Agriculture and Biotechnology, Zhejiang University, Hangzhou 310058, China; 2Group of Vegetable Breeding, Hainan Institute of Zhejiang University, Sanya 572025, China; 3Laboratory of Horticultural Plant Growth & Quality Regulation, Ministry of Agriculture and Rural Affairs, Hangzhou 310058, China

**Keywords:** *Brassica carinata*, Ethiopian mustard, genetic diversity, aphid resistance, *Myzus persicae*, electrical penetration graph assay, glucosinolate

## Abstract

*Brassica carinata* (BBCC, 2n = 34) is commonly known as Ethiopian mustard, Abyssinian mustard, or carinata. Its excellent agronomic traits, including resistance to biotic and abiotic stresses, make it a potential genetic donor for interspecific hybridization. *Myzus persicae* (green peach aphid, GPA) is one of the most harmful pests of *Brassica* crops, significantly effecting the yield and quality. However, few aphid-resistant *Brassica* crop germplasms have been utilized in breeding practices, while the underlying biochemical basis of aphid resistance still remains poorly understood. In this study, we examined the genetic diversity of 75 *B. carinata* accessions and some plant characteristics that potentially contribute to GPA resistance. Initially, the morphological characterization showed abundant diversity in the phenotypic traits, with the dendrogram indicating that the genetic variation of the 75 accessions ranged from 0.66 to 0.98. A population structure analysis revealed that these accessions could be grouped into two main subpopulations and one admixed group, with the majority of accessions (86.67%) clustering in one subpopulation. Subsequently, there were three GPA-resistant *B. carinata* accessions, BC13, BC47, and BC51. The electrical penetration graph (EPG) assay detected resistance factors in the leaf mesophyll tissue and xylem. The result demonstrated that the Ethiopian mustard accessions were susceptible when the phloem probing time, the first probe time, and the G-wave time were 20.51–32.51 min, 26.36–55.54 s, and 36.18–47.84 min, respectively. In contrast, resistance of the Ethiopian mustard accessions was observed with the phloem probing time, the first probe time, and G-wave time of 41.18–70.78 min, 181.07–365.85 s, and 18.03–26.37 min, respectively. In addition, the epidermal characters, leaf anatomical structure, glucosinolate composition, defense-related enzyme activities, and callose deposition were compared between the resistant and susceptible accessions. GPA-resistant accessions had denser longitudinal leaf structure, higher wax content on the leaf surface, higher indole glucosinolate level, increased polyphenol oxidase (PPO) activity, and faster callose deposition than the susceptible accessions. This study validates that inherent physical and chemical barriers are evidently crucial factors in the resistance against GPA infestation. This study not only provide new insights into the biochemical basis of GPA resistance but also highlights the GPA-resistant *B. carinata* germplasm resources for the future accurate genetic improvement of *Brassica* crops.

## 1. Introduction

Ethiopian mustard (*Brassica carinata*, BBCC, 2n = 34) belongs to the family Brassicaceae, originating from the Ethiopian highlands in Northeast Africa. The species has evolved as a result of a few interspecific hybridization events between the wild *B. nigra* (BB, 2n = 16) and *B. oleracea* (CC, 2n = 18) species in Ethiopia [1]. *B. carinata* is an important contributor to the local agricultural production in Africa, with its edible oil and leaves supplementing the human diet [2]. Previous studies showed that *B. carinata* is also involved in the phytoremediation of heavy metals and in the control of soilborne pathogens [3,4]. The *B. carinata* oil can be converted into biodiesel, a superior renewable energy source with high stability [5]. Recent studies indicated that carinata-based aviation fuel could significantly reduce carbon emissions of the aviation sector [6]. In addition, it is rich in erucic acid, which makes it highly desirable for industrial applications, such as in the production of plastics, lubricants, paints, leather tanning, and cosmetics [7].

Long-term breeding success was achieved mainly through the discovery and utilization of germplasm resources with elite genes. *B. carinata* has numerous desirable agronomic traits, including a strong resistance to pod shattering, as well as resistance to various biotic and abiotic stresses, such as black leg, black rot disease, and various pests, for example, aphids and stink bugs [8,9,10,11]. Due to its superior agronomic traits, *B. carinata* has received increased attention from many *Brassica* crop breeders.

A genetic diversity analysis is an important tool for germplasm resource evaluation, conservation, and utilization for breeding new varieties [12]. Molecular markers and their relationship to phenotypes are essential for elucidating genetic variations [13]. Microsatellites, also known as simple sequence repeats (SSRs), short tandem repeats (STRs), or single-sequence-length polymorphisms (SSLPs), are the shortest and most widely distributed simple repeating DNA sequences in the genomes of eukaryotes [14]. The SSR technique is broadly applied in genetic diversity research, population genetics, evolution research, and molecular marker-assisted breeding [15]. Previous reports have shown that *B. carinata* has a narrow genetic diversity [16,17,18]. However, contrasting studies using morphological traits and other molecular approaches detected a relatively wide genetic diversity of *B. carinata* [19,20].

Aphids are pests in 40 plants families, affecting hundreds of species, such as *Brassica* crops and weedy crucifers. They also act as vectors of more than one hundred viral diseases, which has led to severe economic losses [21]. Compared to the chewing insects, aphids have highly specialized stylets that can penetrate the phloem mainly through the parenchyma in an intercellular manner and extract plant photo assimilates without destroying the structural tissues [22]. When aphids penetrate the plant tissue, aphids produce saliva in salivary glands and inject it into plant tissues [23]. The gelatinous saliva and watery saliva are secreted along the stylet pathway through non-phloem tissues. When the aphid stylets reach the phloem, the aphids start to secrete watery saliva into sieve elements [24]. Aphids can regulate the secretion of watery and gelatinous saliva according to the surrounding conditions of its mouthparts [25]. In addition, they control their host plant responses, affect nutrient distribution, and inhibit plant defense responses by feeding and excreting salivary proteins [26,27]. Similar to other plant-related organisms, aphids inject effectors into their hosts to regulate plant cell processes and achieve plant interactions [28]. An electrical penetration graph (EPG) assay is effective for studying the stylet activity, salivation, or aphid food intake [29], and their activity on the plant is recorded as waveforms, which are specific to different probing and feeding behaviors [30]. An EPG assay can be applied to explore the differences in aphid behaviors on resistant and susceptible plants to locate areas of resistance factors in the leaves [31]. However, only a few aphid-resistant germplasms have been used for breeding, and the underlying biochemical basis of aphid resistance in cruciferous crops so far remains poorly understood.

Generally, aphid resistance to plants is categorized as antixenosis, antibiosis, and tolerance [10,32]. In the early stage of pest selection, plant volatiles such as cembratriene-ol (CBT-ol), linalool, and (E)-β-farnesene (EβF) guide or disrupt the olfactory organs of aphids localizing the plant [33]. For mutual compatible interactions between plant and aphid, the latter penetrates the adherent layer of the leaf and sticks its mouthparts into the cells all the way to the phloem and xylem for sap intake. Plants respond to aphid infestation through their inherent physical barriers, such as trichome, thick stem cortex, spines, and waxes [34,35]. Plants also produce toxic chemicals such as terpenoids, alkaloids, phenols, plant lectins, and protease inhibitors to kill or inhibit pest growth [36,37]. Glucosinolates are plant secondary metabolites found exclusively in Brassicaceae plants, with crucial roles in the herbivore–plant and pathogen–plant interactions [38,39]. Aliphatic glucosinolates and indole glucosinolates have different anti-insect activities, and the latter can improve the anti-GPA activity in plants [40,41,42]. Furthermore, plants respond to mechanical damage or microbial invasion by triggering active defense responses, such as callose deposition, reactive oxygen species (ROS) accumulation, hormone concentration changes, and toxin synthesis [39]. Glucosinolates degradation products have broad-spectrum antifungal activity and can be used as signaling molecules to induce callose deposition and activate the innate immune response of plants [38]. Callose deposition is a natural permeable barrier that contributes to phloem occlusion during herbivore damage, thus controlling infestation by the phloem-feeding insects [43,44]. This study explored the genetic diversity of 75 *B. carinata* accessions, and three GPA resistance accessions were used to evaluate the biochemical GPA resistance. The EPG assay located the resistant factors in the leaf mesophyll tissue and xylem. In addition, the underlying biochemical basis of resistance in the selected Ethiopian mustard accessions against GPA infestation was evaluated. These results will lay out a theoretical and technological basis for the future genetic improvement of *Brassica* crops.

## 2. Results

### 2.1. Growth Characters Investigation of the Ethiopian Mustard Accessions

A total of 75 *B. carinata* accessions have been cultivated in the open field at the agricultural experimental station of Zhejiang University (Appendix A). A total of 29 morphological traits were investigated, including 10 quantitative traits (Appendix A) and 19 qualitative traits (Appendix A). The genetic diversity index of the quantitative traits ranged from 0.00 to 1.53, and the coefficient of variation ranged from 0.00% to 51.65%. High diversity in the leaf shape, leaf wax powder, and plant type was observed (Appendix A). The genetic diversity index of the quantitative traits ranged from 1.954 to 2.062, with high diversity in plant width, seed diameter, and plant height being observed (Appendix A). These results indicated that abundant diversity exists among *B. carinata* accessions. The morphological cluster plot mainly divided the tested accessions into three subgroups. The first subgroup contained 46 accessions, with genotypes such as BC10, BC72, BC70, and BC47. The second subgroup contained 27 accessions, such as BC14, BC38, BC57, and BC60, while the third subgroup contained BC41 and BC75 (Appendix A).

It is noteworthy that some of the *B. carinata* accessions used in this study are cultivated at the experimental farm of Tibet Agricultural and Animal Husbandry University (TAAHU) in Nyingchi City, Tibet Autonomous Region (altitude of 2997 m, 94.347595° E/29.673658° N). The time of the life cycle from seed to seed for *B. carinata* accessions in Nyingchi was 150 days, which was about 30 days less than for those planted in Hangzhou, China. In contrast, the biomass of Hangzhou *B. carinata* plants was remarkably larger than that of Nyingchi plants (Appendix A). The result showed that both Nyingchi and Hangzhou areas were suitable for *B. carinata* accessions to complete their life cycle. However, the environmental and climatic conditions of the highland were similar to that of its original Ethiopian highland; thus, the time of the life cycle for *B. carinata* accessions in the highland is remarkably shorter than that of plants in the lower plain areas.

### 2.2. Seed Qualitative Characteristics of the Ethiopian Mustard Accessions

In this study, the seed qualitative characteristics of the 75 Ethiopian mustard accessions were evaluated using the near-infrared reflectance spectroscopy (NIR reflectance spectroscopy). As shown in Appendix A, the oil content of the tested accessions ranged from 24.3% to 41.7%, while the glucosinolate, protein, and water contents ranged from 98.10 to 154.30 µmol/g, 23.20% to 33.55%, and 2.10% to 5.45%, respectively. In addition, the contents of erucic acid, oleic acid, linoleic acid, and saturated fatty acid in the oil ranged from 0 to 25.45%, 55.40% to 70.30%, 4.65% to 11.45%, and 5.40% to 7.35%, respectively (Appendix A). The coefficient of variation in the 75 accessions ranged from 5% to 66% and was relatively low for the oil, glucosinolate, protein, and water contents. In contrast, 66% higher saturated fatty acid and erucic acid contents were observed in the oil extract (Appendix A). Accessions with low erucic acid contents and high oleic acid are potential candidate varieties for edible oil production [45].

### 2.3. Genetic Diversity Analysis of the Ethiopian Mustard Accessions Using SSR Markers

Thirty-eight high polymorphism primers were selected from a collection of 387 SSR primers (Appendix A), and used to obtain 119 strips. Of these, five strips were common and a polymorphic rate of 95.80%. These strips were used for genetic diversity and a population structure analysis.

A relatively wide genetic variation was detected among the tested *B. carinata*, with a genetic similarity (GS) coefficient of between 0.66 and 0.98 (Appendix A). The greatest genetic distance among these accessions was observed in the BC75 genotype from Pakistan, which was consistent with the morphological dendrogram. Except for BC-75, BC-09, BC-32, and BC-03, the other 71 accessions had narrow genetic variations and a GS coefficient of between 0.868 and 0.98 (Appendix A). The log likelihood of *K* in the population structure analysis increased stably with the change in *K* from 1 to 10, and ∆*K* showed a peak at K = 2 (Figure 1A), indicating that the 75 tested accessions could be divided into two main subpopulations. The application of a membership probability threshold at 70% generated three clusters. The SP1 cluster consisted of two (2.67%) accessions, BC75 and BC09, and the SP2 cluster consisted of 65 (86.67%) accessions, while the remaining eight (10.67%) accessions formed an admixture group (AG) (Figure 1C). The phylogenetic tree based on UPGMA classified the 75 accessions in a detailed manner (Figure 1D). The principal component analysis (PCA) also elucidated the relatedness of the tested accessions, and the variance of PC1–PC3 were 11.75%, 8.21%, and 6.14%, respectively. The first two principal components accounted for 19.96% of the total variation. Overall, the PCA, population structure analysis, and phylogenetic analysis results generated consistent results, indicating a high data reliability (Figure 1E).

### 2.4. Screening for Aphid Resistance in the B. carinata Accessions

To understand the aphid resistance of *B. carinata* to select aphid-resistant accessions for the genetic improvement of *Brassica* crops, the aphid resistance of the tested accessions was evaluated through the indoor release of the GPA offspring from one adult female (Appendix A). The BC03, BC04, BC38, BC46, BC50, BC64, and BC69 genotypes were not evaluated in this experiment due to their poor growth conditions. Changes in the aphid population during each accession were recorded at 7, 14, and 21 days after release. The results showed a significant variation in the aphid resistance in the tested Ethiopian mustard accessions (Figure 2), with the aphid number ratio ranging from 0.44 to 1.57 (Appendix A). The BC13, BC18, BC37, BC47, and BC51 genotypes had relatively strong aphid resistance and a low aphid ratio of 0.61 and were designated as moderately resistant (MR) accessions. The aphid number ratio of 20 accessions, such as BC05, BC08, and BC09, ranged between 0.61 and 0.90 and were designated as low-resistant (LR) accessions. The aphid number ratio of 25 accessions, such as BC02, BC06, and BC07, was between 0.91 and 1.20 and were designated as low-susceptibility (LS) accessions. The aphid number ratio of 15 accessions, such as BC19, BC20, and BC22, was between 1.21 and 1.50 and were designated as moderately susceptible (MS) accessions. BC01, BC25, and BC60 expressed poor aphid resistance with aphid number ratios higher than 1.50 and, thus, were designated as highly susceptible (HS) accessions (Figure 2; Appendix A).

### 2.5. EPG Assay of the Leaf Mesophyll and Xylem Resistance Factors for Explaining the Aphid Resistance of Ethiopian Mustard

Aphids have highly specialized piercing-sucking mouthparts that cannot be observed directly in the opaque host tissue during penetration. Therefore, an EPG assay was used to observe their feeding behavior on HS BC01 and resistant BC47 genotypes Appendix A. The results showed that GPA had typical aphid feeding waveforms on *B*. *carinata,* including a nonpenetration phase (np); potential drop (pd); stylet pathway waveforms (A, B, and C); feeding waveforms in the phloem phase (E1 and E2); and xylem phase (G) (Appendix A). The EPG assay was performed after infested GPA, and the feeding wave was analyzed during the feeding process for 6 h. The results indicated that the phloem probing time, the first probe time, and the G-wave time were 20.51–32.51 min, 26.36–55.54 s, and 36.18–47.84 min, respectively, which suggested that the Ethiopian mustard accession were susceptible. In contrast, the phloem probing time, the first probe time and G-wave time values of 41.18–70.78 min, 181.07–365.85 s, and 18.03–26.37 min, respectively, indicated that the Ethiopian mustard accessions were resistant (Table 1). Aphid-feeding behavior varied greatly between the BC01 and BC47 genotypes, particularly in the time and duration of each waveform. An approximately two-fold delay in the time to the first phloem phase was observed in the aphids feeding on BC47 compared to BC01, with an approximately seven-fold longer duration for the first probe and a shorter duration for the total xylem phase. Based on these waveforms, aphids located in the mesophyll and xylem of the Ethiopian mustard encountered physical resistance factors during feeding.

### 2.6. Comparison of Leaf Structure of the Susceptible and Resistant Accessions

Insignificant differences for trichomes, stomatal density, and stomatal size were observed between the susceptible (BC01, BC25, and BC60) and resistant (BC13, BC47, and BC51) genotypes (Appendix A). Comparison of the longitudinal anatomical characteristics revealed that, compared to the susceptible group, the resistant genotypes had thinner leaves, thinner spongy tissues, and denser structures, which partly contributed to their aphid resistance (Figure 3 and Appendix A; Table 2). In addition, the wax content on the leaf surface of the resistant accessions was higher than that of the susceptible accessions (Figure 3).

### 2.7. Aphid Resistance of Ethiopian Mustard Is Related to Its Indole Glucosinolate Content

Significant differences in the composition and content of glucosinolates were observed between the susceptible BC01 and the resistant BC47 genotypes. An overall consistent trend was observed, and the content of indole-3-ylmethylglucosinolate (I3M-GS) was the highest. Compared to BC01, BC47 had a significantly higher content of indole glucosinolates, including I3M-GS, 4MTI3M-GS, and 1MTI3M-GS, but lower content of aliphatic glucosinolates, 3MSOP-GS, 2O3BM-GS, Allyl-GS, and 3BM-GS. Notably, 1MTI3M-GS was only detected in the BC47 genotype (Figure 4). These results suggest that indole glucosinolates contribute to the improved aphid resistance of Ethiopian mustard.

### 2.8. Comparison of Enzymatic Activities between the Susceptible and the Resistant Accession

To further understand the comparative enzymatic activities of POD, PPO, and PAL, regarded as the essential part of the plant defense response to aphids, they were determined between the susceptible BC01 and the resistant BC47 genotypes. After the plants were released by GPA for 48 h, the activity of the PPO enzyme in the resistant accession BC47 was 51.33 U/mg, which was significantly higher than that in BC01 (25.33 U/mg). The PAL activity of resistant material BC47 was 105.00 U/mg, which was significantly lower than that of susceptible material BC01 (248.33 U/mg). However, there was no significant difference in POD activity between the BC01 and BC47 genotypes (Figure 5).

### 2.9. Aphid Resistance of B. carinata Is Associated with Callose Deposition

The callose deposition signal was analyzed at 0, 12, 24, and 72 h after GPA release to determine the different defense strategies of HS BC01, BC25, and BC60 and resistant BC13, BC47, and BC51 genotypes (Figure 6). As a response to the GPA release, the callose deposition signal increased after 72 h of release in both the susceptible and resistant genotypes. The callose deposition signals of the resistant genotypes were remarkably stronger than those of the susceptible genotypes at each time point (Figure 7). These results indicated that callose deposition dramatically increased after aphid release in the resistant genotypes, thus enhancing their capacity to purge GPA feeding.

## 3. Discussion

Evaluation of the genetic diversity in *Brassica* crops using their morphological traits, isozyme activities, and molecular markers can provide the basis for the conservation, development, and utilization of their germplasm resources [12]. An analysis of 75 Ethiopia mustard accessions collected from different countries and regions of Ethiopia showed a wide genetic variability in their morphological characteristics, which was consistent with a previous observation [19]. The centers of crop diversity usually occurred near their region of origin, and numerous plant germplasm resources, such as *B. carinata,* originate from Ethiopia [45]. However, great differences were observed between the morphological cluster plot and dendrogram based on the SSR analysis. This could be as the result of the limited number of morphological traits analyzed in this study, with the environment greatly influencing some of the analyzed traits. In addition, the results of this study indicated that the SSR markers analysis could not replace the morphological analysis in identifying the distinctness of the Ethiopia mustard accessions. Notably, BC75 from Pakistan was successfully separated by both morphological and SSR analyses, which is likely due to the unique environmental and climatic conditions and breeding history of BC75 in Pakistan. The GS dendrogram showed that the GS coefficients of all tested accessions were within the range of 0.66–0.98, which was wider than the previously reported 0.89–0.98 [16] and 0.63–0.88 [17] but smaller than the 0.44–0.87 [20], which was based on an AFLP analysis. The GS coefficients are related to the breadth of the germplasm resource. The reason for the different GS coefficients in different studies may be related to the number of accessions, type of molecular markers, origin of accessions, and so on. Except for the BC75 and BC09 genotypes, the genetic variations of other accessions were relatively narrow, which could be due to their breeding history (especially in Ethiopia) [2,18]. A population structure analysis divided the tested accessions into two subpopulations, with most accessions (96%) falling into one group. Similar results were reported using DArT-seq markers [9] and SNPs [18]. In summary, the genetic diversity of Ethiopian mustard is not abundant compared to cabbage, Chinese cabbage, and mustard, with narrow genetic backgrounds among the accessions.

Evaluation of the seed quality showed that the contents of oil, glucosinolates, and protein in the seeds of tested Ethiopian mustard accessions ranged from 24.3% to 41.7%, 98.10 to 154.30 µmol/g, and 23.20% to 33.55%, respectively, which was consistent with the pervious report [46]. The tested accessions contained an extremely high oleic acid content (55.40–70.30%) and relatively high erucic acid content (0–25.45%). It means these Ethiopian mustard accessions are suitable for breeding as an edible oil crop or biofuel plant [5,46]. Studies on the correlations among oleic acid, linolenic acid, and erucic acid contents in the Ethiopian mustard seeds have previously been reported [47,48]. Consistently, high oleic acid genotype in this study exhibited low erucic acid and linolenic acid contents [19]. The quality of canola oil is mainly determined by its high unsaturated fatty acids contents (oleic acid, linoleic acid, and linolenic acid) and low saturated fatty acids contents. The erucic acid content should not exceed 2% in practice [49]. The tested accessions had high average oleic acid contents and low-to-undetectable erucic acid levels, which made them suitable candidate varieties for edible oil feedstock. In addition, the Ethiopian mustard can overcome the unprecedented effects of global climate change [50].

Due to its excellent agronomic traits, which makes it a potential resource for enhanced edible oil production, similarly, its resistance to biotic and abiotic stresses render it a suitable donor for interspecific hybridization with other *Brassica* species [2,51]. The desirable agronomic traits of Ethiopian mustard can be transferred to five other cultivated *Brassica* species for targeted genetic improvement using the synthetic allohexaploid *Brassica* crop (2n = AABBCC) as a bridge hybrid [17,52].

GPA is an important pest that can destroy the yield and quality of *Brassica* crops and spreads plant viruses in *Brassica* crops and weedy crucifers [53]. Therefore, to analyze the aphid resistance as a stress factor in the 75 Ethiopian mustard accessions, three highly aphid-resistant genotypes, BC13, BC47, and BC51, as well as three HS genotypes, BC01, BC25, and BC60, were evaluated. The results showed that resistant plants were not only highly robust but also had low aphid survival rates with few offspring. These results provide reference for further breeding.

A detailed analysis of the aphid–plant interactions, including aphid-feeding behavior, as well as plant basic and induced resistance mechanisms, can provide useful information for the further utilization of aphid resistance traits. The EPG assay showed significant differences in the time to the first phloem phase, duration of the first probe, and duration of the total xylem phase between the susceptible BC01 and resistant BC47 genotypes. This indicated the inherent occurrence of physical or chemical resistance factors in the surface/epidermis, mesophyll, and xylem of Ethiopian mustard [29]. Other plant–aphid systems also show the importance of the plant epidermis, mesophyll, and xylem resistance factors. For example, *Rhopalosiphum padi* fed on the resistant wild Hsp5 barley delayed in reaching the phloem compared to the susceptible barley Concerto cultivar [54]. Similarly, *R. padi* fed on the resistant maize seedlings exhibited a prolonged penetration time on the epidermis and mesophyll compared to the susceptible maize cultivar [30]. *Aphis glycines* Matsumura fed on the resistant soybean delayed in accessing the phloem compared to the susceptible soybean, and no difference was observed in the duration of phloem sap sucking [55]. These results are consistent with our observation that aphids spend a relatively longer time reaching the host phloem. The feeding difficulty and components of phloem and xylem sap greatly influence the feeding behavior of aphids [35,56]. In general, aphids experience more difficulty accessing the xylem sap than phloem sap due to negative tension [57]. The low aphid survival and reproduction rates on resistant BC47 accession could be attributed to their decreased uptake capacity of the phloem and xylem sap. The *M. persicae* population NL took longer to suck xylem sap on the pepper-resistant cultivar, and NL showed a longer xylem feeding time than the more virulent *M. persicae* population SW, and switching of the NL feeding behavior was possibly to prevent starvation [31], which is in contrast to the current study, as different hosts and aphid populations have their unique forms of interaction.

In terms of leaf surface characteristics, leaf trichome and stomatal density and size contributed minimally to the aphid resistance in Ethiopian mustard. This was confirmed by the EPG assay results that revealed no significant difference in the duration of the non-probing phase (a key indicator of epidermal-mediated resistance) between the BC01 and BC47 genotypes [29,54]. Further studies are still needed on other leaf epidermis features related to aphid resistance, such as volatiles, glandular secretions, and waxes [56]. However, a dense leaf structure has been speculated to be associated with the aphid resistance in Ethiopian mustard by increasing the difficulty of intercellular puncture and reducing the palatability and digestibility of the leaf tissues [22,34,56]. The EPG assay and leaf anatomy results showed that, due to the dense structure of the resistant genotypes, the aphid mouthparts delayed in passing through the mesophyll tissue to reach the xylem and phloem, which prevented them from obtaining nutrients [57].

Glucosinolates are the main defense metabolites of the Brassicaceae species against herbivores and microbial pathogens [38,58]. The aphid resistance in Ethiopian mustard was closely related to its indole glucosinolate content. The resistant BC47 genotype had a higher indole glucosinolate content (I3M-GS, 4MTI3M-GS, and 1MTI3M-GS) than the susceptible BC01 genotype, while the opposite trend was observed for the content of aliphatic glucosinolates. Similar to *Arabidopsis thaliana* and most *Brassica* species, I3M-GS was detected as the predominant indole glucosinolate in Ethiopian mustard. Both I3M-GS and 4MTI3M-GS have good anti-aphid activity [40,42,59,60]. IMT3M-GS was only detected in resistant Ethiopian mustard, suggesting its possible crucial role as an aphid-resistant compound in *B. carinata*; however, this hypothesis needs further clarification. Studies have previously explored the disparate anti-insect activities of indole and aliphatic glucosinolates [42], and the former has been shown to be less stable [59] and is spontaneously activated in the absence of myrosinase. The breakdown products after ingestion inhibit the reproduction of aphids or other crucifer-specific herbivores [42,59]. However, aliphatic glucosinolates can pass intact through the aphid gut [59] and, thus, are mostly negligible and even have beneficial effects on aphids [39]. The anti-aphid effect of indole glucosinolates has comprehensively been studied in *Arabidopsis thaliana*; however, the precise anti-aphid mechanism in other cruciferous plants needs further investigation.

Plant defense-related enzymes peroxidase (POD), polyphenol oxidase (PPO), and phenylalanine ammonia-lyase (PAL) regulate the secondary metabolite levels and participate in the endogenous defense responses of plants to different biotic stresses [61]. During aphid penetration into the leaf epidermis, phenolic compounds are rapidly synthesized and polymerized in the cell wall [62]. In this process, PPO and PAL are the key secondary metabolic enzymes that mediate plant resistance to aphids [32]. Many studies have shown that the increased activity of the PPO enzyme can improve the resistance of tomatoes, peppers, and *Arabidopsis* against aphids [63]. In this study, the level of PPO enzyme activity in the resistant BC47 accession was significantly higher than in the susceptible accessions, indicating that this enzyme might be involved in aphid resistance.

Callose deposition is an important defense response strategy to attack by herbivores and pathogens in plants [43,64] and is an important defense mechanism of wheat, *A. thaliana*, corn, soybeans, and pepper against aphids [44,58,62]. Plant wounds and aphid saliva–protein components can stimulate plant defenses, resulting in callose deposition [28,62] and contributing to the occlusion of the phloem vessels for suppressed phloem sap leakage [65]. Thus, aphids cannot feed on the phloem for long periods. However, susceptible wheat varieties showed increased callose deposition after aphid exposure; callose deposition is a direct wound response mechanism in response to an aphid attack, leading to tolerance [66]. Indole glucosinolate biosynthesis is a likely requirement for the pathogen-induced callose response [38], but the effect of indole glucosinolates on callose-induced deposition by aphids is unknown. In this study, callose deposition was identified as a defense mechanism against aphid attack in Ethiopian mustard. However, production and its role in the Ethiopian mustard, as well as its aphid interaction, need further investigation.

## 4. Materials and Methods

### 4.1. Plant Materials

A total of 75 *B. carinata* accessions were collected from the U.S. National Plant Germplasm System, the North Eastern Plant Intro. Station, Geneva, N. Y. USDA, ARS (USA), The Ethiopian Holetta Agricultural Research Center (Holetta, Ethiopia), Oil Crops Research Institute, Chinese Academy of Agricultural Sciences (Wuhan, China), and the laboratory of Professor Zou Jun (Huazhong Agricultural University, Wuhan, China) (Appendix A). Plant materials were collected from 11 countries, with the majority obtained from Ethiopia.

### 4.2. Evaluation and Statistics of Agronomical Characteristics

The *B. carinata* accessions were cultivated at the Zijingang agricultural experimental station of Zhejiang University in Hangzhou City, Zhejiang Province (altitude of 12 m, 120.085157° E/30.308124° N). Normal field management was performed throughout the life cycle of plants, and their morphological traits in the field were investigated and recorded following the reference descriptors for *Brassica* and *Raphanus* at 40 days after transplantation [67]. Each accession was grown in duplicates in a plot size of 1.2 × 3.0 m^2^. Three randomly tagged plants were sampled per plot per accession after the rosette leaves were fully developed. A total of 29 agronomic traits were investigated, including 19 qualitative and 10 quantitative traits (Appendix A). Microsoft Excel 2016 was used for data processing and analysis. Qualitative traits were standardized (Appendix A), and quantitative traits were divided into 10 grades to eliminate dimensional effects for comparison. SPSS 19.0 was employed to construct the morphological cluster plot using the method of average connection between groups and the Euclidean distance metric.

Grade 1: Xi < X − 2σ, Grade 10: Xi > X + 2σ; each grade differs by 0.5σ [68].

Coefficients of variation (CV) = σ/X.

X = mean value; σ = standard deviation; Xi = measured data.

The Shannon diversity index was used for the genetic diversity analysis as follows:

H′ = −ΣPilnPi (Pi = number of individuals of grade i/total number of individuals) (i: 1–10) [16,69].

### 4.3. Identification of Seed Quality Features using NIR Reflectance Spectroscopy

All *B. carinata* accessions were artificially pollinated at the flowering stage to ensure homozygosity. Whole, intact *B. carinata* seeds were scanned by Foss NIRS systems 5000 (Foss NIR Systems, Copenhagen, Denmark). Approximately 3 g of intact seed samples were placed in a small ring-shaped cup with an inner diameter of 36 mm. The spectral region was 1100–2500 nm with a resolution of 2 nm. WinISI II V1.5 software (Foss NIR Systems, Copenhagen, Denmark) was used to collect the spectra. The average of three repeats was used as the spectral data for a given sample, and 75 averaged spectra were collected corresponding to the 75 samples of *B. carinata* accessions, according to the references [70,71].

### 4.4. DNA Isolation and Genetic Diversity Analysis using SSR Markers

DNA templates were extracted in triplicates from young leaves of each accession by using the modified cetyltrime thylammonium bromide (CTAB) method. DNA template quality was determined by UV spectrophotometry and 1% agarose gel electrophoresis. Five landraces ‘Youqin 49′ (genome = AA, as outgroup), BC-28, BC-31, BC-32, and BC-47 were selected to screen 387 SSR primer pairs (327 primer pairs from *B. rapa* genome and 60 primer pairs from *B. oleracea* genome). Subsequently, 38 SSR primers were screened out to genotype the 75 *B. carinata* accessions (Appendix A). The selected primers were used for the PCR amplification of 75 accessions, and the products were isolated in 12% nondenatured polyacrylamide gel (Native-PAGE). The whole experiment was performed in triplicate.

The PCR analysis with a total reaction (10 μL) included: 37.5 ng of template genomic DNA; 0.25 μL Taq DNA polymerase (Fermentas^TM^); 0.25 μL of (10 μmol·L^−1^) dNTPs; 1 μL (10 μmol/L) of upstream and downstream primers; 1 μL (10 μmol/L) of 10× PCR Buffer (200 mmol/L Tris pH 8.0, 200 mmol/L KCl, 100 mmol/L (NH4)_2_SO_4_, and 20 mmol/L MgSO_4_); and ddH_2_O to 10 μL. The PCR amplification program was as follows: initial denaturation at 94 °C for 3 min, followed by 35 cycles of 30 s denaturation at 94 °C, 30 s annealing at 55 °C, and 30 s extension at 72 °C, with a final extension for 7 min at 72 °C.

SSR data were recorded in an Excel worksheet to form a 0/1 matrix, with 0 and 1 representing absence and presence, respectively. The NTSYS2.10e software and Power Marker V3.25 software were used to analyze the genetic distance and its cluster relationship. STRUCTURE software was employed to run 10 repeats on the 0–1 matrix data with a range of *K* values (1–10) to determine the appropriate subpopulations of 75 accessions [72]. The online tool Structure Harvester (http://taylor0.biology.ucla.edu/struct_harvest/, accessed on 21 August 2021) was applied to obtain the best K and individual and population Q matrix data. Repeat sampling analysis was conducted with CLUMPP1.1.2b. Principal component analysis (PCA) was performed by using Past 3 software [73].

### 4.5. Screening for Aphid Resistance under Controlled Conditions

The 75 *B. carinata* accessions were planted in 10-cm-diameter pots and kept in a standard climate chamber (22 ± 3 °C, 60–70% relative humidity, and L16:D8 photoperiod). Plants were arranged in a randomized block design with five replicates and three blocks.

GPA clones were developed from a single Virginia parous female collected from the laboratory and reared on Chinese cabbage under the same standard conditions above. Seedlings at 15 days were exposed to five GPAs, and nymphs were released in the middle of each plant (Appendix A). Plant growth and aphid number were recorded at 7, 14, and 21 days after release. Aphid number ratio was used as the basis of the resistance assessment, and the previously reported grading standards were applied as a reference [74]:

Aphid number ratio (I/i) = highest aphid number of each tested material (I)/average aphid number of all observed materials (i). I/i> 1.5, highly susceptible (HS); 1.20 < I/i ≤ 1.50, moderately susceptible (MS); 0.90 < I/i ≤ 1.20, lowly susceptible (LS); 0.60 < I/i ≤ 0.90, lowly resistant (LR); 0.30 < I/i ≤ 0.60, moderately resistant; 0.01 < I/i ≤ 0.30, highly resistant; and I/i = 0, immune (I) [75].

### 4.6. Monitoring of GPA Feeding using EPG Assay

The stylet penetration activities of aphids (approximately 7–10 days old, food deprivation for 2 h prior to detection) on different accessions (seedlings with 3–5 true leaves) were monitored by the DC-EPG system (Beijing Channel Scientific Instrument Co., Ltd., Beijing, China) for 6 h at 20 °C ± 4 °C under constant light in the laboratory. The plant probe was inserted into the plant soil, and the aphid probe was connected using a 10–20-μm gold wire and water-based adhesive attached to the aphid’s dorsum (Appendix A). All EPG assay recordings were obtained inside a grounded Faraday cage to limit external noise.

Experimental set-ups for aphids fed on BC01 and BC47 genotypes were prepared in five effective replicates. In each treatment, only aphids that were active for every 6 h of recording were considered as valid replicates. The 6-h feeding statistics of aphids were analyzed and classified using EPG Stylet ana v.21 and EPG Stylet dnd v.19 software.

### 4.7. Scanning Electron Microscopy (SEM)

The sixth true leaf of HS (BC01, BC25, and BC60) and MR (BC13, BC47, and BC51) *B. carinata* accessions were sampled for SEM analysis. Leaf sections (diameter 0.8 cm circular) were fixed (2.5% glutaraldehyde in 0.1 M potassium phosphate buffer, pH 7.0); washed (0.1 M potassium phosphate buffer, pH 7.0, four times, 15 min each); and further fixed (1% osmium acid, 1–2 h) after being dehydrated by successive washes in an ethanol series (30%, 50%, 70%, and 90%, 15 min each) and anhydrous acetone (add anhydrous calcium chloride 2 days in advance, three times, 15 min each). The samples were further dehydrated in HCP-2 (Hitachi, Japan, Tokyo) critical point dryer apparatus, coated with gold in IB-5 (EIKO, Japan, Tokyo) Sputter Coater, and viewed with a XL-30 ESEM (Philips, Eindhoven, The Netherlands). SEM images were observed and counted in Adobe Photoshop CS6 and Excel 2016.

### 4.8. Histological Observation

The sixth true leaves of HS (BC01, BC25, and BC60) and MR (BC13, BC47, and BC51) *B. carinata* accessions were sampled for semi thin section analysis. Leaf sections (1.0 cm by 0.5 cm) were fixed (2.5% glutaraldehyde in 0.1 M potassium phosphate buffer, pH 7.0), washed (0.1 M potassium phosphate buffer, pH 7.0, four times, 15 min each), and further fixed (1% osmium acid, 1–2 h) after being dehydrated by successive washes in an ethanol series (30%, 50%, 70%, 80%, 90%, 95%, and 100%, 15 min each). After being transferred to anhydrous acetone for 30 min, the samples were treated with a mixture of EPON812 epoxy resin and anhydrous acetone (*v*/*v* = 1/1) for 1 h and then with a mixture of EPON812 epoxy resin and anhydrous acetone (*v*/*v* = 3/1) for 3 h. The samples were subsequently transferred to a pure embedding agent Epon812 epoxy resin overnight at room temperature, then embedding repeated the next day to pure embedding agent Epon812 epoxy resin again overnight at 70 °C. Sections were observed on a Leica RM2255 (Leica, Weztlar, Germany) microtome under a light microscope (Nikon 90i).

### 4.9. Analysis of Glucosinolate Content

Glucosinolate was extracted as previously described [76], with minor modifications. Briefly, 100 mg freeze-dried samples of seedling leaf (samples from BC01 and BC47 genotypes) were boiled in 1 mL of water for 10 min and then centrifuged at 7000× *g* for 5 min, then collected supernatant. The residues were boiled once again with 1 mL of water and then centrifuged at 7000× *g* (5 min). The aqueous extract of two times was combined and then transferred to a DEAE-Sephadex A-25 (30 mg) column (pyridine acetate form) (GE Healthcare, Detroit, MI, USA), which was washed three times with 1 mL of pyridine acetate (20 mM) and then twice with 1 mL of water twice. Desulphoglucosinolates were obtained by eluting with 1 mL of water after the overnight treatment of 100 μL of 0.1% aryl sulphatase (Sigma, St. Louis, MI, USA) at RT.

The HPLC analysis of desulphoglucosinolates was performed on a Shimadzu HPLC instrument (Shimadzu, Kawasaki, Japan, Kawasaki) equipped with SPD-M20A diode array detector (Shimadzu, Kawasaki, Japan). The samples (20 μL) were separated on a Hypersil C18 column (5 μm, 4.6 mm × 250 mm) (Elite Analytical Instruments Co., Ltd., China, Shanghai) using acetonitrile and water at a flow rate of 1.0 mL/min. Absorbance was detected at 226 nm. The mobile phases were as follows: 1.5% acetonitrile for 5 min, a linear gradient to 20% acetonitrile for the next 15 min, and isocratic elution with 20% acetonitrile over the final 10 min. Sinigrin (Sigma, St. Louis, USA) was used as internal standard for HPLC analysis.

The glucosinolates were qualitatively analyzed by LC-MS. The samples (50 μL) were separated on a Prontosil ODS2 C18 column (5 μm, 4.6 mm × 250 mm) using acetonitrile and water at a flow rate of 1.0 mL/min. Absorbance was detected at 227 nm. The mobile phases were as follows: a linear gradient 0% to 20% acetonitrile for the first 20 min and keep 20% acetonitrile over the next 15 min. MS parameters were as follows: ion source: ESI(+); sprayer pressure: 60 psi; dry gas (N2) flow rate: 13 L/min; dry temperature: 350 °C; capillary voltage: 4000 V; fragment voltage: 100 V; and scanning range: 100.00–600.00 m/z. Data were collected by Agilent 1100 LC/MSD (Agilent, Paro Alto, CA, USA) chemical workstation, and Mass Hunter Qualitative Analysis B.07.00 was used to analyze the glucosinolate component in different accessions.

### 4.10. Measurement of Defense-Related Enzyme Activities

Young leaves of the susceptible BC01 and the resistant BC47 genotypes, were sampled and crushed into powder with liquid nitrogen after 48 h of GPA release. The activities of POD (peroxidase, EC 1.11.1.7), PPO (polyphenol oxidase, EC 1.10.3.1) and PAL (phenylalanine ammonia-lyase, EC 4.3.1.5) have been determined. About 0.5 g leaf sample were homogenized with 5-mL enzyme extract buffer (pH 6.8) and centrifuged at 12,000× *g* at 4 °C for 30 min. The supernatant was taken as the enzyme extract. A 3.2 mL assay mixture was used for POD activity consisted of 2 mL 0.1 M phosphate buffer (pH 5.5), 1 mL 0.25% guaiacol solution, 0.1 mL enzyme extract and 0.1 mL 0.75% H_2_O_2_ solution. Fully mixed the mixture and incubated at 37 °C for 10 min. The sample was then transferred into boiling water for 5 min. One unit of POD activity was defined as the amount of enzyme required to cause a change of 0.01 in the OD_470_ per minute (U/mg protein). PPO were extracted from 0.2 g leaf sample in 5 mL 0.05 mg/L phosphate buffer (pH 6.5) and centrifuged at 12,000× *g* at 4 °C for 20 min. The supernatant was taken as the enzyme extract. A 5 mL assay mixture was used for PPO activity consisted of 1 mL 0.1 M cataphile, 3.9 mL 50 mM PBS (pH6.5) and 0.1 mL enzyme extract. Then, the treated mixture was mixed well and incubated at 30 °C for 10 min. One unit of PPO activity was defined as the amount of enzyme required to cause a change of 0.01 in the OD_525_ per minute (U/mg protein). PAL was extracted from 0.2 g leaf sample in 5 mL precooled 0.1 M boric acid-borax buffer (pH8.7, containing 1 mM EDTA, 0.1 g PVP, and 20 mM β-mercaptoethanol) and centrifuged at 12,000× *g* at 4 °C for 20 min. A 5 mL assay mixture was used for PAL activity which consisted of 1 mL 0.6 M phenylalanine, 3.9 mL extraction buffer and 0.1 mL enzyme extract. Then, the treated mixture was mixed well and incubated at 37 °C for 1 h, 0.2 mL 6 M hydrochloric acid was added to terminate the reaction. One unit of PAL activity was defined as the amount of enzyme required to cause a change of 0.01 in the OD_290_ per minute (U/mg protein).

### 4.11. Analysis of Callose Deposition

Seedlings of HS (BC01, BC25, and BC60) and MR (BC13, BC47, and BC51) genotypes with 3–5 true leaves were prepared for aphid feeding treatment. The two sides of the main vein of leaf in the same position were fixed with tiny insect cages. Ten aphids (approximately 7–10 days old) were placed in each insect cage. Leaf samples were taken at 0, 12, 24, and 72 h after release. Each treatment was conducted in triplicates. Frozen section and aniline blue staining were used to observe callose deposition. The samples were fixed in 30% sucrose solution overnight at 4 °C and transferred first to a mixture of 30% sucrose solution and OCT (Tissue OCT-Freeze Medium) (*v*/*v* = 1/1) solution for 12 h at 4 °C, then to pure OCT overnight at 4 °C, and finally to pure OCT at −80 °C. Sections with a thickness of 14 μm were prepared on Shandon FE + Shandon Finnesse32 (Thermo, USA) freezing slicer at −20 °C and then stained for 1.5 h in 1% aniline blue in the dark. After staining, callose deposition was observed and photographed under a fluorescence microscope. Images were analyzed with Image-Pro Plus 6.0, and area of callose deposition was calculated as a percentage of the total area.

For statistical analysis, all values were expressed as the mean ± SEM. Data were further subjected to ANOVA, and means were compared using the least significant difference (LSD) test (SPSS 19.0). *p* < 0.05 was considered statistically significant.

## 5. Conclusions

In summary, this study showed that the genetic variation of 75 tested *B. carinata* accessions ranged from 0.66 to 0.98 and could be grouped into two main subpopulations with an admixture group. Subsequently, three accessions (BC13, BC47, and BC51) were selected as aphid-resistant accessions for further study. GPA-resistant factors were located in the *B. carinata* leaf mesophyll tissue and xylem by EPA assay. Three feeding parameters, the phloem probing time, the first probe time, and the G-wave time, have significant differences between the susceptible and resistant accessions. Aphid-resistant accessions showed a denser leaf longitudinal structure, higher wax content on the leaf surface, higher indole glucosinolate level, increased PPO enzyme activity, and faster callose deposition than the susceptible accessions. This study validates that inherent physical and chemical barriers are evidently involved in the resistance against GPA infestation.

## Figures and Tables

**Figure 1 ijms-23-13736-f001:**
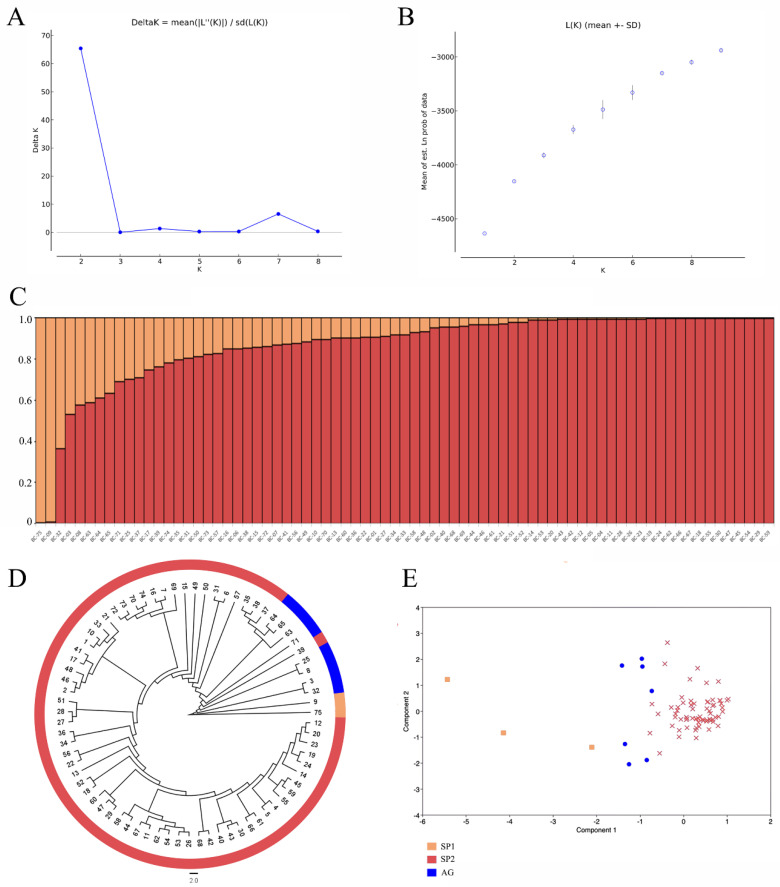
Population structure analysis and PCA of the 75 Ethiopian mustard accessions. (**A**) Estimation of the optimum number of groups (K). (**B**) Graph for the parameter L(K) and number of K. (**C**) Population structure at K = 2. (**D**) Cluster analysis of the 75 Ethiopian mustard accessions. Numerical values 1–75 represent accessions BC01–BC75. (**E**) PCA of the 75 Ethiopian mustard accessions.

**Figure 2 ijms-23-13736-f002:**
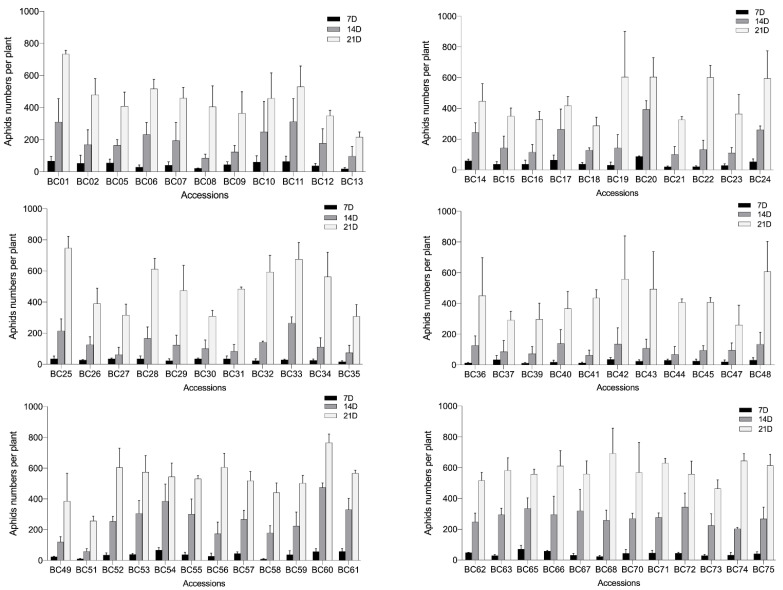
Variations in aphid numbers among the 75 Ethiopian mustard accessions at 7, 14, and 21 days after infestation. Bars represent the means ± SE.

**Figure 3 ijms-23-13736-f003:**
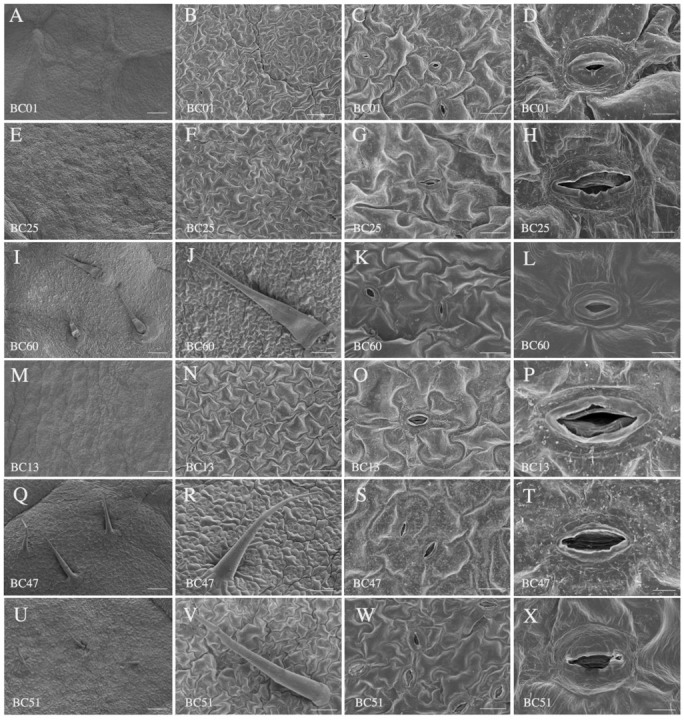
Characteristics of lower leaf epidermis of the susceptible and the resistant accessions. (**A**–**L**) Leaf lower epidermis characteristics of the susceptible accessions BC01, BC13, and BC60. (**M**–**X**) Leaf lower epidermis characteristics of resistant accessions BC13, BC47, and BC51. Scale bars: 250 μm in (**A**,**E**,**I**,**M**,**Q**,**U**), 100 μm in (**B**,**F**,**J**,**N**,**R**,**V**), 20 μm in (**C**,**G**,**K**,**O**,**S**,**W**), and 5 μm in (**D**,**H**,**L**,**P**,**T**,**X**).

**Figure 4 ijms-23-13736-f004:**
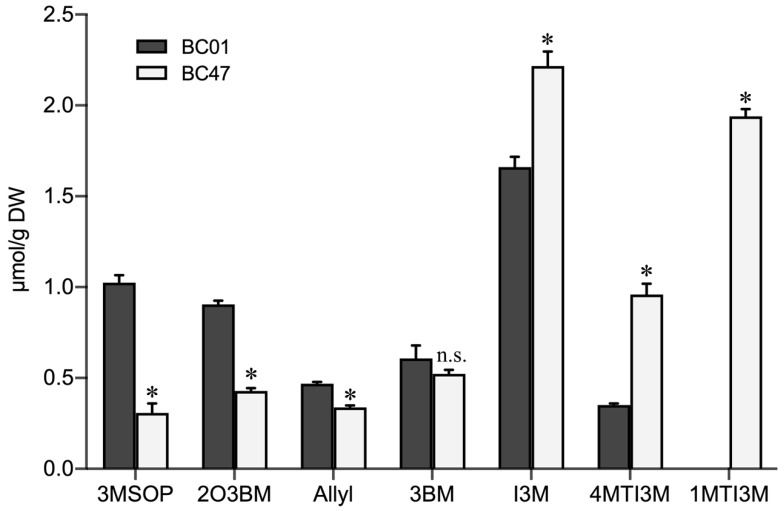
Comparison of glucosinolates compositions and contents in the susceptible and resistant Ethiopian mustard leaves. Notes: 3MSOP: 3-methyl propyl glucosinolates; 2O3BM: 2-hydroxy-3-butenyl glucosinolates; Allyl: 2-allyl glucosinolates: 3BM:3-butylene glucosinolates; I3M: 3-indole methyl glucosinolate; 4MTI3M: 4-methoxy-3 indole glucosinolates; 1MTI3M: 1-methoxy-3 indole glucosinolates. Bars are the means ± SE with three biological replicates. Asterisks indicates statistically significant values from those of BC01 (Fisher’s LSD test; *p* < 0.05); n.s.: no statistical significance between BC01 and BC47.

**Figure 5 ijms-23-13736-f005:**
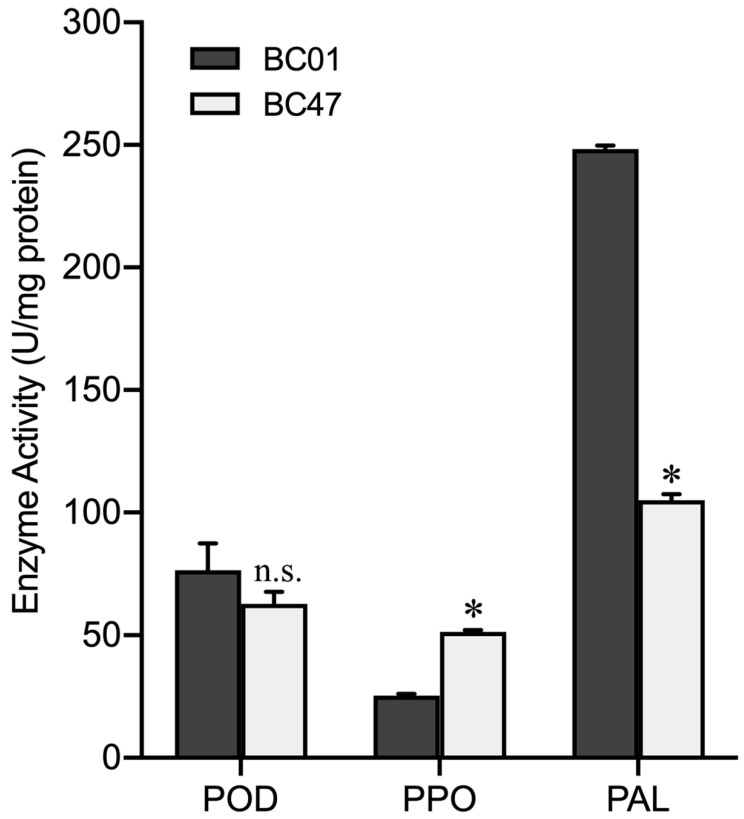
Activities of three defense-related enzymes in the susceptible and resistant Ethiopian mustard leaves. Notes: Bars are the means ± SE with three biological replicates. Asterisks indicate statistically significant values from those of BC01 (Fisher’s LSD test; *p* < 0.05); n.s.: no statistical significance between BC01 and BC47.

**Figure 6 ijms-23-13736-f006:**
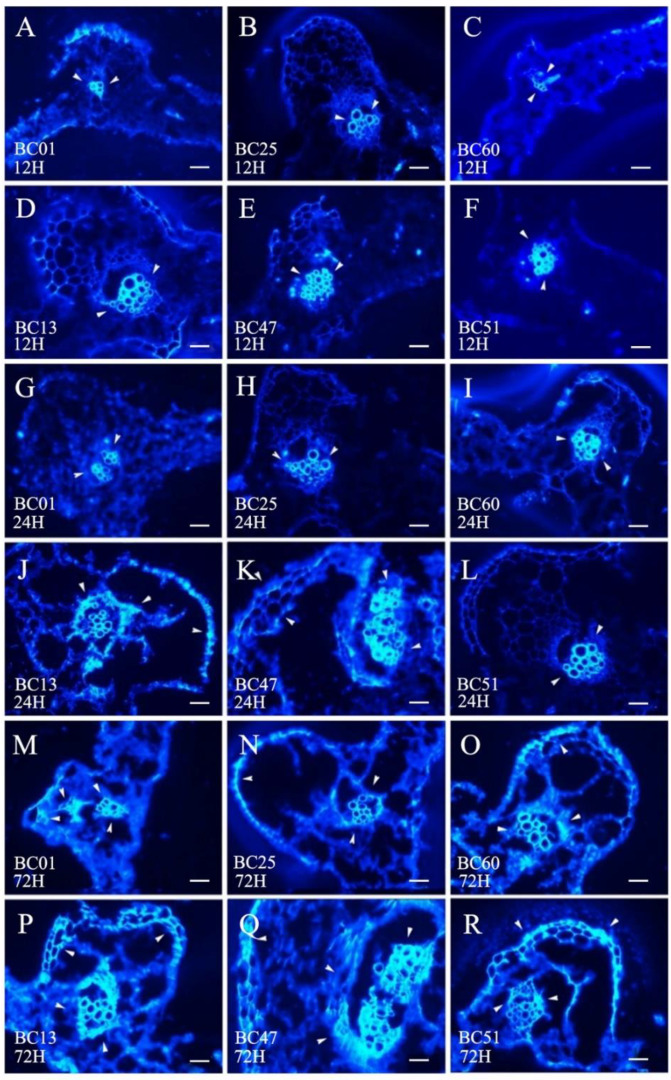
Comparison of callose deposition between the susceptible and the resistant Ethiopian mustards at different time points after infestation with GPA. Notes: (**A**–**F**) are 12 h after infestation with GPA between three susceptible accessions (BC01, BC25, and BC60) and three resistant accessions (BC13, BC47, and BC51); (**G**–**L**) are 24 h after infestation with GPA between the three susceptible accessions (BC01, BC25, and BC60) and three resistant accessions (BC13, BC47, and BC51); (**M**–**R**) are 72 h after infestation with GPA between the three susceptible accessions (BC01, BC25, and BC60) and three resistant accessions (BC13, BC47, and BC51). The white arrows indicate the callose deposition position. Scale bars = 100 μm.

**Figure 7 ijms-23-13736-f007:**
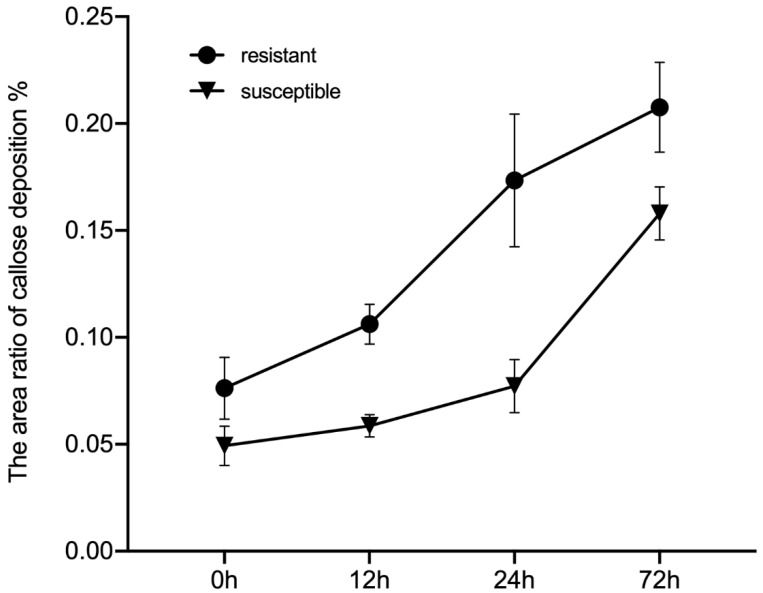
Observed callose deposition between the susceptible and the resistant Ethiopian mustards at different time points after infestation with GPA. Notes: BC01, BC25, and BC60 are the susceptible accessions, and BC13, BC47, and BC51 are the resistant accessions. The white arrows indicate the callose deposition position. Scale bars = 100 μm.

**Table 1 ijms-23-13736-t001:** The feeding behaviors of aphids on the susceptible and resistant Ethiopia mustard accessions.

EPG Parameters	Mean Value ± Standard Error of the Mean	
	Susceptible (BC01)	Resistant (BC47)
Number of probes/(*n*)	208.00 ± 23.07	224.67 ± 32.53
Time to first phloem phase (E1 and E2)/min	26.51 ± 6.00	55.98 ± 14.80 *
Duration of first probe/s	40.95 ± 14.59	273.46 ± 92.39 *
Duration of total np/min	83.57 ± 14.62	76.92 ± 21.90
Duration of total probe/min	179.70 ± 12.00	221.76 ± 19.49
Duration of total xylem phase (G)/min	42.01 ± 5.83	22.20 ± 4.17 *
Duration of total phloem phase (E1 and E2)/min	49.16 ± 10.94	39.12 ± 7.83

Notes: Total recorded time is 6 h, Five effective duplications for each treatment. Values represent the means ± SE. Significant differences were compared at 0.05 and 0.01 according to the Mann–Whitney *U* test (* *p* < 0.05).

**Table 2 ijms-23-13736-t002:** Characteristics and parameters of the lower leaf epidermis and leaf structure of susceptible and resistant accessions.

Characteristics	Mean Value ± Standard Error of the Mean					
	Susceptible Genotypes			Resistant Genotypes		
	BC01	BC25	BC60	BC13	BC47	BC51
Trichomes height (μm)	Null	Null	565.82 ± 27.20 Ns	Null	538.03 ± 16.89 Ns	425.26 ± 20.55 Ns
Trichomes density (mm^−2^)	Null	Null	565.82 ± 27.20 Ns	Null	1.27 ± 0.12 Ns	1.38 ± 0.38 Ns
Stomatal size (μm)	4.75 ± 0.74 a	16.45 ± 0.51 b	420.29 ± 14.50 b	16.45 ± 0.51 b	13.91 ± 0.33 b	15.47 ± 0.43 b
Stomatal density (mm^−2^)	338.16 ± 22.13 ab	205.04 ± 3.71 ab	11.06 ± 0.90 b	144.93 ± 14.50 a	178.41 ± 22.57 ab	258.06 ± 16.13 ab
Leaf thickness (μm)	313.23 ± 15.08 a	291.17 ± 18.76 a	290.27 ± 8.78 a	214.73 ± 4.29 b	234.17 ± 4.47 b	272.53 ± 7.27 ab
Palisade mesophyll thickness (μm)	196.63 ± 15.52 a	180.60 ± 5.50 a	159.80 ± 2.51 ab	135.23 ± 2.75 b	137.63 ± 3.66 b	157.57 ± 11.76 a
Palisade mesophyll (layer)	4.33 ± 0.58 ab	4.33 ± 0.58 ab	4.00 ± 0.02 b	5.67 ± 0.58 a	5.67 ± 0.58 a	6.33 ± 0.58 a
Spongy mesophyll thickness (μm)	101.43 ± 3.23 a	99.40 ± 2.93 a	97.50 ± 3.82 a	67.00 ± 4.16 b	56.87 ± 1.84 b	64.13 ± 2.39 b
Spongy mesophyll (layer)	5.00 ± 1.00 a	4.33 ± 0.58 a	4.67 ± 0.58 a	3.67 ± 0.58 a	2.67 ± 0.58 a	2.67 ± 0.58 a
Leaf thickness (μm)	313.23 ± 15.08 a	291.17 ± 18.76 a	290.27 ± 8.78 a	214.73 ± 4.29 b	234.17 ± 4.47 b	272.53 ± 7.27 ab

Note: Values represent the means ± SE, and different alphabetical letters are significant at *p* < 0.05 (Fisher’s LSD test).

## Data Availability

The data presented in this study are available in the article. Additional data are available on request from the corresponding author.

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
