# Peer review of "Genetic Diversity Analysis Reveals Potential of the Green Peach Aphid (Myzus persicae) Resistance in Ethiopian Mustard"

_ijms, 2022, doi:10.3390/ijms232213736_

Round 1

Reviewer 1 Report

The study reports the resistance to biotic and abiotic stresses of Ethiopian mustard (Brassica carinata). The authors believe that it is an excellent donor for interspecific hybridization. Brassica carinata is well-known mustard among plant breeders.  The manuscript has four main sections; I) Morphological characterization II) Seed morphological and content characterization, III) Resistance against Myzus persicae (GPA) and IV) Enzyme and some content activity such as glucosinolate which are related to resistance. Behind the manuscript has hard work. It is very clear. The underlying questions refer to has Brassica carinata rich genetic variations? And, do these different variations, have some resistance character against GPA? These questions are answered in the text and documented. the manuscript has several shortcomings that make it unclear, especially work on GPA growth rate on accessions. What do you mean by same growth status!!, If you mean stage, you should call it "nymph". Which stage was used you should indicate. If you look at your S7 table, BC13 and BC37 are MR but the population growth on day 7 is quite different, and please look the day 14, it has also speculating your theory too, why is it happening??

The growth rate in most of the accessions has a similar wave!!!! If it is resistant, it is resistant on day 1 or seven or 14 too. BC31 and BC37 show this wave very clearly!!!. This table is the center of your resistance work.

More specific points are listed below.

1.            Table and figure legends (including supplementary tables have to be written with more and clear details ?

2.            literature 21 (Page 2 paragraph 4)  is not related to the vector activity of Myzus persicae, and this aphid transmits more than 100 viruses.

3.            Page 3 the title 2.1, second paragraph is not related to the title “Morphological characterization of Ethiopian mustard accessions”

4.            Page 3 title 2.2 started with a discussion, please carry this paragraph to the discussion.

5.            Page 6 first graph in Figure 6, Y-axis has to be 1000 like the others

6.            Figure 1 D.

                The manuscript has minor problems. One of the major questions is whether Ethiopian mustard has rich genetic diversity or not. According to title 2.3 (page 4) except for BC-75, BC-09, BC-32, and BC-03 it had narrow genetic variations. But in the discussion, the author said that “Ethiopia mustard accessions collected from different countries and different regions of Ethiopian showed enormous genetic variability for morphological characteristics”. These two findings deserve better discussions.

Author Response

Response letter for reviewers’ comments

The response letter to reviewers as follows:

Reviewer 1

The study reports the resistance to biotic and abiotic stresses of Ethiopian mustard (Brassica carinata). The authors believe that it is an excellent donor for interspecific hybridization. Brassica carinata is well-known mustard among plant breeders.  The manuscript has four main sections; I) Morphological characterization II) Seed morphological and content characterization, III) Resistance against Myzus persicae (GPA) and IV) Enzyme and some content activity such as glucosinolate which are related to resistance. Behind the manuscript has hard work. It is very clear. The underlying questions refer to has Brassica carinata rich genetic variations? And, do these different variations, have some resistance character against GPA? These questions are answered in the text and documented. the manuscript has several shortcomings that make it unclear, especially work on GPA growth rate on accessions. What do you mean by same growth status!!, If you mean stage, you should call it "nymph". Which stage was used you should indicate. If you look at your S7 table, BC13 and BC37 are MR but the population growth on day 7 is quite different, and please look the day 14, it has also speculating your theory too, why is it happening??

The growth rate in most of the accessions has a similar wave!!!! If it is resistant, it is resistant on day 1 or seven or 14 too. BC31 and BC37 show this wave very clearly!!!. This table is the center of your resistance work.

Answer: Thanks for your valuable comment. In this study, the 75 Ethiopia mustard accessions collected from different countries and different regions of Ethiopian showed enormous genetic variability for morphological characteristics, and this finding is consistent with the previous study (Alemayehu and Becker,2002). Based on the results of morphological characteristics investigation in this study, there are rich genetic variability such as leaf type, leaf color, leaf shape, petiole color, and so on. However, in this study, GS dendrogram of SSR markers showed that the GS coefficients of the 75 accessions were within the range of 0.66–0.98, which was wider than that in the study of Warwick (Warwick et al., 2006) (0.89–0.98) and Jiang (Jiang et al., 2007) (0.63–0.88) but constricted than that in the research of Teklewold and Becker (0.44–0.87) based on AFLP data research (Teklewold and Becker, 2006). The reason for the different GS coefficients in different research may be related to the number of materials, type of molecular marker and origin of accessions, and so on.

To investigate the genetic diversity of the two diploid Brassica species, cabbage and Chinese cabbage, fifty-four B. oleracea cultivars, including 14 broccoli (B. oleracea var. italica), 13 cauliflflower (B. oleracea ar. botrytis), and 27 cabbage (B. oleracea var. capitata) cultivars from different sources were evaluated for SSR polymorphisms. Results indicated the similarity values ranged from 0.54 to 0.96 for cabbage cultivars, 0.75 to 0.94 for broccoli cultivars, and 0.82 to 1.0 for cauliflflower cultivars(Tonguc and Griffiths, 2004). Subsequently, the core SNPs were used for construction of a neighbour‐joining dendrogram that separated the 70 Chinese cabbage inbred lines into four main groups and several subgroups corresponding to Caixin, Heiyebaicai, Huangxinwu, Naibaicai, Taitsai, Pak‐choi, and Wutatsai. Furthermore, the polymorphic information content (PIC) of SSR markers in Pak‐choi (0.31) was lower than that of core SNP markers in non‐heading Chinese cabbage (0.34) (Li et al., 2019). Thus, among the Brassica crops of U’ triangle, there are not rich genetic variations in B. carinata and B. nigra based on the morphyological investigation and molecular markers screening, however, there are lot of variations in B. rapa, B. oleracea, B. napus and B. juncea, such as leafy heads, enlarged organs (roots, stems and inflorescences) and extensive axillary branching. Some of the best examples of this phenomenon include the leafy heads of Chinese cabbage (B. rapa) and cabbage (B. oleracea) and root or stem tubers in turnip (B. rapa), kohlrabi (B. oleracea), swede (B. napus) and tuberous mustard (B. juncea)(Chen et al., 2016).

In summary, the genetic diversity of Ethiopian mustard is not abundant compared to cabbage, Chinese cabbage and mustard, with narrow genetic backgrounds among accessions. We have revised our related expression in text. Subsequently, we have also revised “the same growth status” with “nymph” in text.

In addition, about the growth rate of Ethiopian mustard accessions, results showed that most of them had the same wave. The aphid resistance of Ethiopian mustard depend on inherent physical barrier, volatiles and inhibiting factors of plants, and there is interaction between plants and aphids(Powell et al., 2006). Thus, we compared the aphid numbers changes for 21 days when we carried out the experiment of aphid resistance screening under controlled conditions (Table S7). Further analysis of the aphid resistance screening results indicated that the population growth of BC13 and BC37 is quite different on day 7, and the final evaluation result is quite different accordingly. Furthermore, the population growth of BC31 and BC37 are almost same level on day 7 and 14, and the final evaluation result is also quite different. Result shows that the aphid resistance of Ethiopian mustard is interrelated with the growth stage of plant. However, the population growth of BC56 and BC66 on day 7 is quite different, but the final evaluation result is almost same. These results indicated the aphid resistance of Ethiopian mustard were quite complicated and needed to be long-term experiment. It is unclear for the underlying biological mechanism of aphid resistance in Ethiopian mustard to date, and a better understanding of these events may lead to improved management strategies.

 More specific points are listed below.

  1. Table and figure legends (including supplementary tables have to be written with more and clear details ?

Answer: Thanks for your valuable comment. The table and figure legends (including supplementary tables and figures) have revised and make them more clear.

  1. literature 21 (Page 2 paragraph 4)  is not related to the vector activity of Myzus persicae, and this aphid transmits more than 100 viruses.

Answer: Thanks for your valuable comment. We have used another reference (Brault, V., Uzest, M., Monsion, B., Jacquot, E., Blanc, S. Aphids as transport devices for plant viruses. C. R. Biol. 2010, 333: 524-538. https://doi:10.1016/j.crvi.2010.04.001) to substitute the reference 21.

  1. Page 3 the title 2.1, second paragraph is not related to the title “Morphological characterization of Ethiopian mustard accessions”

Answer: Thanks for your valuable comment. The title 2.1 has been revised as “Growth characters investigation of Ethiopian mustard accessions”.

  1. Page 3 title 2.2 started with a discussion, please carry this paragraph to the discussion.

Answer: Thanks for your valuable comment. In title 2.2, we carried this paragraph to the discussion. In addition, other titles started with a discussion expressions have been revised at the same time.

  1. Page 6 first graph in Figure 2, Y-axis has to be 1000 like the others

Answer: Thanks for your valuable comment. In Figure 2, the Y-axis has revised as 1000 like the others.

  1. Figure 1 D.

Answer: I am sorry, I don’t know what’s problem in Figure 1 D, because the reviewer do not refer to it.

  1. The English requires improvement. I strongly recommend that you employ a professional editor with knowledge of the subject area.

Answer: Thanks for your valuable comments. The manuscript has been revised by the professional company, and the certificate is showed as follow.

Reference:

  1. Alemayehu, N., Becker, H.C. Genotypic diversity and patterns of variation in a germplasm material of Ethiopian mustard (Brassica carinata A. Braun). Genet. Res. and Crop Evol. 2002, 49, 573-582. https://doi.org/10.1023/A:1021204412404.
  2. Cheng, F., Sun, R., Hou, X., et al. Subgenome parallel selection is associated with morphotype diversification and convergent crop domestication in Brassica rapa and Brassica oleracea. Nature genet. 2016, 48(10): 1218. https://doi:10.1038/ng.3634
  3. Jiang, Y., Tian, E., Li, R., Chen, L., Meng, J. Genetic diversity of Brassica carinata with emphasis on the interspecific crossability with B. rapa. Plant Breeding 2007, 126, 487-491. https://doi.org/10.1111/j.1439-0523.2007.013.
  4. Li, P., Su, T., Wang, H., Zhao, X., Wang, W., Yu, Y., Zhang, D., Wen, C., Yu, S., Zhang, F. Development of a core set of KASP markers for assaying genetic diversity in Brassica rapa subsp. chinensis Makino. Plant Breed. 2019, 138: 309-324. https://doi.org /10.1111/pbr.12686
  5. Powell, G, Tosh, C. R., Hardie, J. Host plant selection by aphids: behavioral, evolutionary, and applied perspectives. Annu. Rev. Entomol. 2006, 51:309-30. https:// doi:10.1146/annurev.ento.51.110104.151107
  6. Teklewold, A., Becker, H.C. Geographic pattern of genetic diversity among 43 Ethiopian mustard (Brassica carinata A. Braun) accessions as revealed by RAPD analysis. Genet. Res. and Crop Evol. 2006, 53, 1173-1185. https://doi.org/10.1007/ s10722-005-2011-4.

7.Tonguc, M., Griffiths, P.D. Genetic relationships of Brassica vegetables determined using database derived simple sequence repeats. Euphytica 2004, 137: 193-201. https://doi.org/10.1023/B:EUPH.0000041577.84388.43

  1. Warwick, S.I., Gugel, R.K., McDonald, T., Falk, K.C. Genetic variation of Ethiopian mustard (Brassica carinata A. Braun) germplasm in western Canada. Genet. Res. and Crop Evol. 2006, 53, 297-312. https://doi.org/10.1007/s10722-004-6108.

Special thanks to you for your good comments, these comments can help us to improve our manuscript significantly. We really benefit a lot. Thank you!

Best regards to you,

Xiaolin YU

Reviewer 2 Report

Clarify the duration of the experiment as for resistance study one year data may not be sufficient because there may be chances of escape which will mislead the resistance level.

In conclusion, how the author concluded high gene flow between only two subpopulations, 

Author Response

Response letter for reviewers’ comments

The response letter to reviewers as follows:

Reviewer 2

  1. The manuscript has minor problems. One of the major questions is whether Ethiopian mustard has rich genetic diversity or not. According to title 2.3 (page 4) except for BC-75, BC-09, BC-32, and BC-03 it had narrow genetic variations. But in the discussion, the author said that “Ethiopia mustard accessions collected from different countries and different regions of Ethiopian showed enormous genetic variability for morphological characteristics”. These two findings deserve better discussions.

Answer: Thanks for your valuable comment. In this study, the 75 Ethiopia mustard accessions collected from different countries and different regions of Ethiopian showed enormous genetic variability for morphological characteristics, and this finding is consistent with the previous study (Alemayehu and Becker,2002). Based on the results of morphological characteristics investigation in this study, there are enormous genetic variability such as leaf type, leaf color, leaf shape, petiole color, and so on. However, in this study, GS dendrogram of SSR markers showed that the GS coefficients of the 75 accessions were within the range of 0.66–0.98, which was wider than that in the study of Warwick (Warwick et al., 2006) (0.89–0.98) and Jiang (Jiang et al., 2007) (0.63–0.88) but constricted than that in the research of Teklewold and Becker (0.44–0.87) based on AFLP data research (Teklewold and Becker, 2006). The reason for the different GS coefficients in different research may be related to the number of materials, type of molecular marker and origin of accessions, and so on.

To investigate the genetic diversity of the two diploid Brassica species, cabbage and Chinese cabbage, fifty-four B. oleracea cultivars, including 14 broccoli (B. oleracea var. italica), 13 cauliflflower (B. oleracea ar. botrytis), and 27 cabbage (B. oleracea var. capitata) cultivars from different sources were evaluated for SSR polymorphisms. Results indicated the similarity values ranged from 0.54 to 0.96 for cabbage cultivars, 0.75 to 0.94 for broccoli cultivars, and 0.82 to 1.0 for cauliflflower cultivars(Tonguc and Griffiths, 2004). Subsequently, the core SNPs were used for construction of a neighbour‐joining dendrogram that separated the 70 Chinese cabbage inbred lines into four main groups and several subgroups corresponding to Caixin, Heiyebaicai, Huangxinwu, Naibaicai, Taitsai, Pak‐choi, and Wutatsai. Furthermore, the polymorphic information content (PIC) of SSR markers in Pak‐choi (0.31) was lower than that of core SNP markers in non‐heading Chinese cabbage (0.34) (Li et al., 2019). Thus, among the Brassica crops of U’ triangle, there are not rich genetic variations in B. carinata and B. nigra based on the morphyological investigation and molecular markers screening, however, there are lot of variations in B. rapa, B. oleracea, B. napus and B. juncea, such as leafy heads, enlarged organs (roots, stems and inflorescences) and extensive axillary branching. Some of the best examples of this phenomenon include the leafy heads of Chinese cabbage (B. rapa) and cabbage (B. oleracea) and root or stem tubers in turnip (B. rapa), kohlrabi (B. oleracea), swede (B. napus) and tuberous mustard (B. juncea)(Chen et al., 2016).

In summary, the genetic diversity of Ethiopian mustard is not abundant compared to cabbage, Chinese cabbage and mustard, with narrow genetic backgrounds among accessions.

Meanwhile, in the present study, it is worth mentioning that the BC75 can successfully be separated by both of morphological and SSR analyses. The probable reasons are that genetic mutation due to the enviromental and climatic changes and foreign genes introgression of BC75 in Pakistan are unique. The process of domestication and selection of a crop plays a decisive role in its genome composition. The BC-09, BC-32, and BC-03 originate from Ethopia three provinces, respectively, where is the original centre of Ethopian mustard. In accordance with previous suggestions that Ethiopia is the primary centre of origin for B. carinata and the species has spread across diferent continents through migration with early human civilization. Thus, it is worth carrying out the further investigation to explore the reason of excepting for the three accessions result in the GS coefficients reduction significantly in the near future.

  1. Clarify the duration of the experiment as for resistance study one year data may not be sufficient because there may be chances of escape which will mislead the resistance level.

Answer: Thanks for your valuable comment. It is true, several years data is better to the duration of the experiment as for resistance study. In this study, in order to mislead the resistance level, the testing materials were arranged in randomized block design with five replicates and three blocks. According to the previous studies, results showed aphid population development was largely independent of the under glasshouse and field conditions, but the trend was almost similar during two years or the two environmental conditions, results indicated consistency of the resistance responses over two years on an aphid population basis (Broekgaarden et al., 2008; Atri et al., 2012). Some studies with one year data have also made good progress (Kumar et al., 2011; Hao et al., 2019). Subsequently, in this study, the results of EPG assay and leaf anatomical structure can reasonably explain the aphid resistance of the BC47 accession.

  1. In conclusion, how the author concluded high gene flow between only two subpopulations.

Answer: Thanks for your valuable comment. Based on the cluster result in Fig. 1 C, we found that about half of the accessions had some kind of genomic intermingling, but it was just a matter of magnitude. A recent study provided a comprehensive analysis of diversity among B. carinata germplasm available from resource centres worldwide, and identifed thousands of genome-wide SNPs using GBS. The diversity observed suggests B. carinata originated from a very limited number, if not a single hybridization event, with little or no subsequent inter-specifc crossing with the parental progenitors (Khedikar et al., 2020). After comprehensive consideration, we have removed this sentence in abstract, text and conclusion.

  1. The English requires improvement. I strongly recommend that you employ a professional editor with knowledge of the subject area.

Answer: Thanks for your valuable comments. The manuscript has been revised by the professional company, and the certificate is showed as follow.

Reference:

  1. Alemayehu, N., Becker, H.C. Genotypic diversity and patterns of variation in a germplasm material of Ethiopian mustard (Brassica carinata A. Braun). Genet. Res. and Crop Evol. 2002, 49, 573-582. https://doi.org/10.1023/A:1021204412404.
  2. Atri, C., Kumar, B., Kumar, H., Kumar, S., Sharma, S., Banga, S.S. Development and characterization of Brassica juncea – fruticulosa introgression lines exhibiting resistance to mustard aphid (Lipaphis erysimi Kalt). BMC Genet. 2012, 13: 104. http:// doi:10.1186/1471-2156-13-104
  3. Broekgaarden, C., Poelman, E.H., Steenhuis, G., Voorprips, R.E., Dicke, M., Vosman, B. Responses of Brassica oleracea cultivars to infestation by the aphid Brevicoryne brassicae: an ecological and molecular approach. Plant, Cell Environ. 2008, 31: 1592-1605. http:// doi: 10.1111/j.1365-3040.2008.01871.x
  4. Cheng, F., Sun, R., Hou, X., et al. Subgenome parallel selection is associated with morphotype diversification and convergent crop domestication in Brassica rapa and Brassica oleracea. Nature genet. 2016, 48(10): 1218. https://doi:10.1038/ng.3634
  5. Hao, Z.P., Zhan, H.X., Wang, Y.L., Hou, S.M. How cabbage aphids Brevicoryne brassicae (L.) make a choice to feed on Brassica napus cultivars. Insects 2019, 10, 75. http://doi:10.3390/insects10030075
  6. Jiang, Y., Tian, E., Li, R., Chen, L., Meng, J. Genetic diversity of Brassica carinata with emphasis on the interspecific crossability with B. rapa. Plant Breeding 2007, 126, 487-491. https://doi.org/10.1111/j.1439-0523.2007.013.
  7. Khedikar, Y., Clarke, W.E., Chen, L.F., Higgins, E.E., Kagale, S., Koh, C.S., Bennett, R., Parkin, I.A.P. Narrow genetic base shapes population structure and linkage disequilibrium in an industrial oilseed crop, Brassica carinata A. Braun. Sci. Rep. 2020, 10, 12629. https://doi.org/10. 1038/s41598-020-69255.
  8. Kumar S., Atri, C., Sangha, M.K., Banga, S.S. Screening of wild crucifers for resistance to mustard aphid, Lipaphis erysimi (Kaltenbach) and attempt at introgression of resistance gene(s) from Brassica fruticulosa to Brassica juncea. Euphytica 2011, 179: 461-470. http:// doi:10.1007/s10681-011-0351-z
  9. Li, P., Su, T., Wang, H., Zhao, X., Wang, W., Yu, Y., Zhang, D., Wen, C., Yu, S., Zhang, F. Development of a core set of KASP markers for assaying genetic diversity in Brassica rapa subsp. chinensis Makino. Plant Breed. 2019, 138: 309-324. https://doi.org /10.1111/pbr.12686
  10. Teklewold, A., Becker, H.C. Geographic pattern of genetic diversity among 43 Ethiopian mustard (Brassica carinata A. Braun) accessions as revealed by RAPD analysis. Genet. Res. and Crop Evol. 2006, 53, 1173-1185. https://doi.org/10.1007/ s10722-005-2011-4.

11.Tonguc, M., Griffiths, P.D. Genetic relationships of Brassica vegetables determined using database derived simple sequence repeats. Euphytica 2004, 137: 193-201. https://doi.org/10.1023/B:EUPH.0000041577.84388.43

  1. Warwick, S.I., Gugel, R.K., McDonald, T., Falk, K.C. Genetic variation of Ethiopian mustard (Brassica carinata A. Braun) germplasm in western Canada. Genet. Res. and Crop Evol. 2006, 53, 297-312. https://doi.org/10.1007/s10722-004-6108.

Special thanks to you for your good comments, these comments can help us to improve our manuscript significantly. We really benefit a lot. Thank you!

Best regards to you,

Xiaolin YU

Reviewer 3 Report

The submitted manuscript “Genetic diversity analysis reveals potential of the green peach aphid (Myzus persicaeresistance in Ethiopian mustard”  by Fangyuan Zhou , Chaoquan Chen , Lijun Kong , Shenglanjia Liu , Kun Zhao , Yi Zhang , Tong Zhao , Kaiwen Liu , Xiaolin Yu presents a very broad study on  the potential resistance in Brassica carinata against Myzus persicae. It attempts to cover many aspects of susceptibility/resistance of B. carinata. The Authors put a lot of work into the study but they have not avoided several inconsistencies and grammatical errors which need correction.

Introduction:

1.       “They also act as vector of more than one hundred viral diseases that led to severe economic losses” – Should be: They also act as vectors of more than one hundred viral diseases that lead to severe economic losses

2.       “When their stylets penetrate through the plant tissue, aphids inject saliva from their feeding places into plant tissues” – Should be: When their stylets penetrate through the plant tissue, aphids inject saliva from their feeding places into plant tissues.

Aphids produce saliva in salivary glands and inject it into plant tissues.

3.       The statement: “First, watery saliva is continuously injected into the phloem, and then gelatinous saliva is injected around the path where the stylet traveled” is not true. The gelling saliva and watery saliva are secreted along the stylet pathway through non-phloem tissues. When aphid stylets reach phloem, aphids start to secrete watery saliva into sieve elements.

4.       „In early stage selection” – what selection? Plant selection by aphids?

Results:

1.       Numbering of the tables and figures should be consecutive and appear successively in the text. Here, the tables S2 and S4 and figure S3 are cited first. Figure S1 appears as late as in the subsection 2.4. and Table S1 – in section 4.1.

2.       Results contain elements of the Discussion or Introduction, i.e., the references to literature appear. The Results should report only the data.

3.       Section 2.4.: What is “purified GPA”? GPA is green peach aphid, but what does ‘purified’ mean?

4.       Section 4.6: what “diet treatment” were the aphids given prior to the EPG experiment?

5.       Why the EPG experiment was done on only two accessions? Why these accessions were chosen?

6.       There are far too few EPG variables analyzed. There are many more variables that have to be used to interpret the resistance/susceptibility properly.

7.       There is no comparison of aphid development on any control plant. The evaluation of resistance is only relative to other studied accessions. The same refers to the EPG experiments. There is no control experiment, for example on Chinese cabbage. At the same time, the variables which are commented as important in susceptible/resistant accessions do not show significant differences. Also, the duration of the first probe on ‘resistant’ plants is almost eight times longer than on susceptible plants. This discrepancy is especially visible in the Abstract. The conclusions are not supported by the results. The Authors should analyze more variables and compare more accessions and perform the control experiment.

Author Response

Response letter for reviewers’ comments

The response letter to reviewers as follows:

Reviewer 3

  1. The submitted manuscript “Genetic diversity analysis reveals potential of the green peach aphid (Myzus persicae)resistance in Ethiopian mustard”  by Fangyuan Zhou , Chaoquan Chen , Lijun Kong , Shenglanjia Liu , Kun Zhao , Yi Zhang , Tong Zhao , Kaiwen Liu , Xiaolin Yu presents a very broad study on  the potential resistance in Brassica carinata against Myzus persicae. It attempts to cover many aspects of susceptibility/resistance of B. carinata. The Authors put a lot of work into the study but they have not avoided several inconsistencies and grammatical errors which need correction.

 Introduction:

  1. “They also act as vector of more than one hundred viral diseases that led to severe economic losses” – Should be: They also act as vectors of more than one hundred viral diseases that lead to severe economic losses

Answer: Thanks for your valuable comment. We have revised as your mentioning in text.

2.“When their stylets penetrate through the plant tissue, aphids inject saliva from their feeding places into plant tissues” – Should be: When their stylets penetrate through the plant tissue, aphids inject saliva from their feeding places into plant tissues.

Aphids produce saliva in salivary glands and inject it into plant tissues.

Answer: Thanks for your valuable comment. We have revised as your mentioning in text.

  1. The statement: “First, watery saliva is continuously injected into the phloem, and then gelatinous saliva is injected around the path where the stylet traveled” is not true. The gelling saliva and watery saliva are secreted along the stylet pathway through non-phloem tissues. When aphid stylets reach phloem, aphids start to secrete watery saliva into sieve elements.

Answer: Thanks for your valuable comment. We have revised as your mentioning in text.

  1. In early stage selection” – what selection? Plant selection by aphids?

Answer: Thanks for your valuable comment. This sentence means that aphids choose plants that they can infect. Thus, we revised this sentence as “ In the early stage of pest selection”.

Results:

  1. Numbering of the tables and figures should be consecutive and appear successively in the text. Here, the tables S2 and S4 and figure S3 are cited first. Figure S1 appears as late as in the subsection 2.4. and Table S1 – in section 4.1.

Answer: Thanks for your valuable comment. We have revised the sequence of supplemental tables and figures according to their appearing successively in the text.

  1. Results contain elements of the Discussion or Introduction, i.e., the references to literature appear. The Results should report only the data.

Answer: Thanks for your valuable comment. We have moved thes paragraphs with a discussion expressions to the part of discussion.

  1. Section 2.4.: What is “purified GPA”? GPA is green peach aphid, but what does ‘purified’ mean?

Answer: Thanks for your valuable comment. The “purified GPA” means that GPA clones were developed from a single virginiparous female collected from the laboratory and reared on Chinese cabbage under the same culture conditions for the 75 B. carinata accessions. We have revised this as “the GPA offsprings come from one adult female”.

  1. Section 4.6: what “diet treatment” were the aphids given prior to the EPG experiment?

Answer: Thanks for your valuable comment. “diet treatment” means food deprivation.Thus, we revised “2 h of diet treatment”as “food deprivation for 2 h”.

  1. Why the EPG experiment was done on only two accessions? Why these accessions were chosen?

Answer: Thanks for your valuable comment. In fact, based on the germination rate of seeds, and the amount of experimental work, the BC01 and BC47 have been selected to be further investigation. It is better to carry out the further investigation with three susceptible lines and three MR lines.

  1. There are far too few EPG variables analyzed. There are many more variables that have to be used to interpret the resistance/susceptibility properly.

Answer: Thanks for your valuable comment. In this study, the results showed that the GPA had typical aphid feeding waveforms on B. carinata including non-penetration phase (np), potential drop (pd), stylet pathway waveforms (A, B, and C), feeding waveforms in phloem phase (E1 and E2), and xylem phase (G) (Table1, Fig. S8). The EPG assay has been performed after infested GPA and analyzing feeding wave during the feeding process for 6 hours. Totally, seven major EPG variables have been analyzed, and three variables out of them have significant difference. These results are almost consistent with the previous study of Myzus persicae on an aphid resistant pepper accession(Sun et al., 2018), and the result of cowpea aphid resistance in cowpea line CB77(MacWilliams et al., 2022). Some studies with seven or eight variables analysis data have also made good progress (Khan et al., 2015; Chen et al., 2018).

  1. There is no comparison of aphid development on any control plant. The evaluation of resistance is only relative to other studied accessions. The same refers to the EPG experiments. There is no control experiment, for example on Chinese cabbage. At the same time, the variables which are commented as important in susceptible/resistant accessions do not show significant differences. Also, the duration of the first probe on ‘resistant’ plants is almost eight times longer than on susceptible plants. This discrepancy is especially visible in the Abstract. The conclusions are not supported by the results. The Authors should analyze more variables and compare more accessions and perform the control experiment.

Answer: Thanks for your valuable comment. In the present study, in the part of title 4.5, “GPA clones were developed from a single virginiparous female collected from the laboratory and reared on Chinese cabbage under the same culture conditions for the 75 B. carinata accessions.” In fact, the aphid-fed Chinese cabbage can be regarded as a control plant (Fig. S1 A). Unfortunately, we did not count the aphid number on 7, 14 and 21 days after the aphid nymph inoculation. Based on visual inspection, the aphid number ratio of Chinese cabbage is roughly equivalent to that of Ethopian mustard accession with moderately susceptible character(following figure).

The aphids (Myzus persicae) reared on Chinese cabbage. a: Adult aphid and nymphs; b:The aphid for inoculation.

In this study, detailed research on aphid–plant interactions, including aphid-feeding behavior and plant constitutive and induced resistance mechanism, can further contribute to utilization of aphid resistance traits. In EPG assay, significant differences were observed in the time to first phloem phase, duration of first probe, and duration of total xylem phase between susceptible material BC01 and resistant material BC47. This finding indicated that physical or chemical resistance factors exist in the surface/epidermis, mesophyll, and xylem of Ethiopian mustard (Alvarez et al., 2006). Other plant–aphid systems also direct the importance of plant epidermis, mesophyll, and xylem resistance factors. Rhopalosiphum padi fed on resistant wild barley Hsp5 spend more time in reaching the phloem than a susceptible barley cultivar (Concerto) (Leybourne et al., 2019).

In addition, in order to facilitate readers to better understand the results of this study, “Three feeding parameters, the phloem probing time, the first probe time and the G-wave time, have significant differences between the susceptible and resistant accessions.” This sentence has been added in the conclusions.

  1. The English requires improvement. I strongly recommend that you employ a professional editor with knowledge of the subject area.

Answer: Thanks for your valuable comments. The manuscript has been revised by the professional company, and the certificate is showed as follow.

Reference:

  1. Alvarez, A.E., Tjallingii, W.F., Garzo, E., Vleeshouwers, V., Dicke, M., Vosman, B. Location of resistance factors in the leaves of potato and wild tuber-bearing Solanum species to the aphid Myzus persicae. Entom. Exp. et Appl. 2006, 121,145-157. https://doi.org/ 10.1111/j.1570-8703.2006.00464.
  2. Chen, C., Ye, S., Hu, H.J., Xue, C.M., Yu, X.P. Use of electrical penetration graphs (EPG) and quantitative PCR to evaluate the relationship between feeding behaviour and Pandora neoaphidis infection levels in green peach aphid, Myzus persicae. J. Insect Physiol. 2018, 104: 9-14. https://doi.org/10.1016/j.jinsphys.2017.11.003
  3. Khan, S. A., Marimuthu, M., Predeech C., Aguirre-rojas L.M., Reese J.C., Smith C. M. Electrical penetration graph recording of Russian wheat aphid (Hemiptera: Aphididae) feeding on aphid-resistant wheat and barley J. Econ. Entomol. 2015, 108(5): 2465-2470. http://doi: 10.1093/jee/tov183
  4. Leybourne, D.J., Valentine, T.A., Robertson, J.A.H., Perez-Fernandez, E., Main, A.M., Karley, A.J., Bos, J.I.B. Defence gene expression and phloem quality contribute to mesophyll and phloem resistance to aphids in wild barley. J. Exp. Bot. 2019, 70, 4011-4026. https://doi.org/10. 1093/jxb/erz163.
  5. MacWilliams, J.R., Chesnais, Q., Nabity, P., Mauck, K., Kaloshian, I. Cowpea aphid resistance in cowpea line CB 77 functions primarily through antibiosis and eliminates phytotoxic symptoms of aphid feeding. J. Pest Sci. 2022, https://doi.org /10.1007/s10340-022-01529-w
  6. Sun, M.J., Voorrips, R. E., Steenhuis-Broers, G., van’t Westende, W., Vosman, B. Reduced phloem uptake of Myzus persicae on an aphid resistant pepper accession. BMC Plant Biol. 2018, 18: 138. https://doi.org/10.1186/s12870-018-1340-3

Special thanks to you for your good comments, these comments can help us to improve our manuscript significantly. We really benefit a lot. Thank you!

Best regards to you,

Xiaolin YU
